# Beneficial Effects of Alpha-Lipoic Acid on Hypertension, Visceral Obesity, UCP-1 Expression and Oxidative Stress in Zucker Diabetic Fatty Rats

**DOI:** 10.3390/antiox8120648

**Published:** 2019-12-16

**Authors:** Adil El Midaoui, I. George Fantus, Ali Ait Boughrous, Réjean Couture

**Affiliations:** 1Department of Pharmacology and Physiology, Faculty of Medicine, Université de Montréal, Montréal, QC H3C 3J7, Canada; rejean.couture@umontreal.ca; 2Research Team “Biology, Environment and Health”, Department of Biology, Faculty of Sciences and Techniques Errachidia, Moulay Ismail University of Meknes, 52000 Errachidia, Morocco; boughrous@gmail.com; 3Department of Medicine, Division of Endocrinology and Metabolism, McGill University Health Centre Research Institute, McGill University, Montreal, QC H4A3J1, Canada; george.fantus@mcgill.ca

**Keywords:** type 2 diabetes, metabolic syndrome, insulin resistance, visceral obesity, hypertension, oxidative stress, α-lipoic acid

## Abstract

Evidence suggests that oxidative stress plays a major role in the development of metabolic syndrome. This study aims to investigate whether α-lipoic acid (LA), a potent antioxidant, could exert beneficial outcomes in Zucker diabetic fatty (ZDF) rats. Male 6-week-old ZDF rats and their lean counterparts (ZL) were fed for six weeks with a standard diet or a chow diet supplemented with LA (1 g/kg feed). At 12 weeks of age, ZDF rats exhibited an increase in systolic blood pressure, epididymal fat weight per body weight, hyperglycemia, hyperinsulinemia, insulin resistance (HOMA index), adipocyte hypertrophy and a rise in basal superoxide anion (O_2_•^−^) production in gastrocnemius muscle and a downregulation of epididymal uncoupled protein-1 (UCP-1) protein staining. Treatment with LA prevented the development of hypertension, the rise in whole body weight and O_2_•^−^ production in gastrocnemius muscle, but failed to affect insulin resistance, hyperglycemia and hyperinsulinemia in ZDF rats. LA treatment resulted in a noticeable increase of pancreatic weight and a further adipocyte hypertrophy, along with a decrease in epididymal fat weight per body weight ratio associated with an upregulation of epididymal UCP-1 protein staining in ZDF rats. These findings suggest that LA was efficacious in preventing the development of hypertension, which could be related to its antioxidant properties. The anti-visceral obesity effect of LA appears to be mediated by its antioxidant properties and the induction of UCP-1 protein at the adipose tissue level in ZDF rats. Disorders of glucose metabolism appear, however, to be mediated by other unrelated mechanisms in this model of metabolic syndrome.

## 1. Introduction

Metabolic syndrome represents a cluster of metabolic abnormalities that comprise central obesity, insulin resistance, hypertension and dyslipidemia. It is increasingly evident that oxidative stress plays a crucial role in the occurrence of metabolic syndrome. In fact, elevated free radicals have been postulated to participate in the pathogenic mechanism and development of complications in insulin resistance, diabetes, hypertension and adiposity [1,2,3,4]. Oxidative stress results from an increased production of reactive oxygen species (ROS) and/or a decreased antioxidant reserve. It has been shown that mitochondria may be one of the predominant sources of endogenous ROS [3]. Superoxide anion formation and NADPH oxidase activity are elevated in the aorta of hypertensive insulino-resistant rats [5]. Likewise, Zucker diabetic fatty (ZDF) rats, a model of type 2 diabetes, exhibit hypertension and obesity accompanied by increased superoxide anion production in the aorta and adipose tissue [6,7].

Uncoupled protein-1 (UCP-1) is a transporter present in the inner membrane of mitochondria that plays a key role in non-shivering thermogenesis. In experimental murine models, brown adipose tissue (BAT) transplantation has positive effects on whole body metabolism and energy homeostasis [8], reinforcing the new concept that increasing BAT mass improves metabolic efficiency. Developing strategies to enhance the brown phenotype in white adipocytes to induce thermogenic activation has been proposed to combat obesity and type 2 diabetes [9,10]. Indeed, studies have shown that anti-diabetic peroxisome proliferator-activated receptor gamma (PPARγ) agonists induce a white-to-brown fat conversion and thermogenic gene expression through activation of UCP-1 [11,12,13]. Interestingly, Lapa et al. [14] have reported that 13-weeks ZDF rats presented ubiquitous white adipose-like tissue, showing large unilocular lipid droplets with a marked decrease in immunostaining mitochondrial concentrations, as well as in immunostaining and protein expression of UCP-1.

Alpha lipoic acid (LA) is a disulfide free-radical scavenger found normally in mitochondria as the coenzyme for pyruvate dehydrogenase and α-ketoglutarate dehydrogenase. Studies have reported that the most abundant vegetable sources of lipoic acid are spinach, broccoli and tomatoes, which contain 3.2, 0.9 and 0.6 × 10^−1^ g lipolyzing/g dry weight, respectively [15,16]. After dietary assimilation, LA is transported into cells and reduced to dihydrolipoic acid (DHLA), which has preeminent antioxidant activity [16]. The α-LA/DHLA oxidation–reduction couple displays a redox potential that is higher than that of vitamin E (α-tocopherol), vitamin C (L-ascorbic acid), ubiquinone (coenzyme Q) and glutathione [17,18]. Thus, α-LA/DHLA has the ability to regenerate the reduced form of vitamin E, vitamin C, coenzyme Q and glutathione, thereby preserving the endogenous reduced state and neutralizing oxidative stress and so further affords a favourable role for this unique antioxidant in multiple physiologic systems either in health and pathology [16,17]. Previous studies have shown that LA prevents the development of hypertension, the enhanced heart mitochondrial superoxide anion production and insulin resistance in chronically glucose-fed rats [19].Moreover, LA was found to improve insulin sensitivity in animal models of insulin resistance and obesity [20,21], to reduce weight gain and to improve the lipid profile in high fat diet treated rats [22]. Studies have shown that treatment of pancreatic islets from type 2 diabetic patients with an antioxidant agent (metformin) can improve beta-cell function and survival [23]. Interestingly, we have shown previously that LA induced PPAR-gamma and PPAR-alpha protein expression in cardiovascular tissues and liver, respectively, in a model of hypertensive insulin-resistant rats [24,25].

This study was designed, therefore, to investigate whether a dietary supplementation of LA could prevent hypertension, insulin resistance, the increase in O_2_•^−^ production in skeletal muscle, adiposity and the alteration in visceral UCP-1 protein expression in ZDF rats, a model of type 2 diabetes.

## 2. Materials and Methods

### 2.1. Ethics Statement

All animal care and experimental procedures complied with the Use of Laboratory Animals and were approved by the University of Toronto (former affiliation of the two first authors) and the Université de Montréal’s Committees on Ethics in accordance with the guiding principles for animal experimentation as enunciated by the Canadian Council on Animal Care. The ethical protocol code was AUP 1123.16 at the Toronto General Hospital Research Institute.

### 2.2. Animals and Protocols

This investigation was undertaken in the well-established animal model of metabolic syndrome, Zucker diabetic fatty (ZDF) rats. Studies were performed in 6-week-old obese male ZDF (fa/fa) rats and age-matched lean male wild-type (fa/+) littermates (ZL) (Charles River Laboratories, St-Constant, QC, Canada). We have chosen 6 weeks because at this age ZDF rats did not yet present the metabolic syndrome characteristics [26]. The rats were divided into four groups of eight rats. A group of ZDF rats and a group of ZL rats were given a standard laboratory chow diet (SD) during 6 weeks, and a group of ZDF rats and a group of ZL rats were fed a SD supplemented with α-lipoic acid (LA, 1 g/kg feed) in the diet during six weeks. The SD and α-lipoic acid supplemented diet were obtained from Ren’s Feed Supplies Limited (Oakville, ON, Canada). All rats had free access to tap water. After six weeks of diet treatment, rats were sacrificed by decapitation under light anesthesia with CO_2_. Subsequently, the blood was collected from the open thorax into a vacutainer tube for the measurements of metabolic parameters. All blood samples were collected early in the morning after 16 h fasting. Organs and tissues were sampled, frozen in liquid nitrogen and stored at −80 °C as described earlier in the same animals for other analysis [7]. Systolic blood pressure was performed by tail-cuff plethysmography and the final value was determined by the calculation of the average of five readings per animal (AD Instruments Inc., Colorado Springs, CO, USA) and was recorded with the AD Instruments Program (Lab Chart Pro7. Ink, Colorado Springs, CO, USA) as conducted in a previous study [27].

### 2.3. Measurement of Metabolic Parameters

Plasma concentrations of glucose and insulin, and the Homeostasis Model Assessment (HOMA) index of insulin resistance, were measured in our previous paper [7].

### 2.4. Measurement of Superoxide Anion

Reactive oxygen species production from the gastrocnemius muscle was evaluated by the lucigenin chemiluminescence method as detailed in our previous paper [7], a well-validated technique for measurement of cells superoxide anion (O_2_•^−^) production [28]. Data were expressed as counts per minute per mg of weight tissue.

### 2.5. Adipocyte Morphometry

Adipocyte morphometry was performed according to our previous studies [27,29]. Briefly, epididymal fat was embedded in paraffin after fixation for 16 h in 4% paraformaldehyde. Then, 20-μm sections were made, deparaffinized and stained with methylene blue. Section snapshots were taken with a DAGE MTI CCD-72 digital camera (Michigan City, IN, USA) and analyzed with MCID software (Imaging Research, Waterloo, ON Canada). Adipocyte cell size (10^–3^ mm^2^) was the average of 20 cells/section (5 sections/rat in 5–6 rats per group). Adipocyte cell number per mm^2^ was the average of 20–30 mm^2^/section (5 sections/rat in 5–6 rats per group). A camera (Q Imaging Retiga-2000R, Surrey, BC, Canada) was used to obtain color photomicrographs.

### 2.6. UCP-1 Protein Immuno-Staining

Epididymal fat tissue sections were made as described for adipocyte morphometry, deparaffinized, rehydrated and incubated in sodium citrate buffer (95 °C, 45 min). Sections were cooled down for 20 min, washed (3 times × 5 min) with 0.1 M PBS buffer (pH 7.4) and incubated in blocking buffer (PBS containing 10% BSA, 10% goat serum and 0.3% triton X-100) for 1 h at room temperature (RT). After overnight incubation at RT, sections were treated with the blocking buffer containing the rabbit anti-UCP1 antibody (1:500, ab10983, Abcam, Cambridge, MA, USA). On the subsequent day, slides were washed (3 × 5 min) in 0.1M PBS and incubated with Alexa Fluor 488 donkey secondary anti-rabbit IgG (1:200, A21206, Thermo Fisher Scientific, Saint-Laurent, QC, Canada) for 2 h at RT. After washing, slides were mounted with ProLong^®^ Gold Antifade Reagent (Thermo Fisher Scientific, Saint-Laurent, QC, Canada. Microphotographs were obtained with a fluorescence microscope (Leica Microsystems Inc., Concord, ON, Canada) and transferred to a computer and analyzed using NIH Image J 1.36b Software (NIH, Bethesda, MD, USA). The camera setting was identical for acquisition of images from all sections. The ratio of the number of pixels for the marker on the number of pixels for the total surface of the analysed image is represented in percentage. Data represent the average of five sections per rat in 5–6 rats per group.

### 2.7. Statistical Analysis

Results are the mean ± standard error of the mean (SEM) of values obtained from 8 rats if not stated otherwise. Statistical analysis was conducted by using Prism^TM^ version 5.0 (GraphPad Software Inc., La Jolla, CA, USA). Data were analysed with one-way ANOVA, followed by the Bonferroni/Dunn multiple comparison test when *F* reached *p* < 0.05 and there was no significant variance in homogeneity. A difference was considered statistically significant when *p* < 0.05.

## 3. Results

### 3.1. Blood Pressure

At 12 weeks of age, ZDF rats showed higher levels of systolic blood pressure (*p* < 0.05; Figure 1) in comparison to ZL rats. The supplementation of the diet with LA had no impact on systolic arterial pressure in ZL rats, yet it prevented the rise in blood pressure in ZDF rats (Figure 1).

### 3.2. Body Weight in Relation to the Weight of Epididymis and Pancreas

The final body weight was significantly higher in ZDF rats (*p* < 0.01; Table 1) in comparison to ZL rats. The supplementation of the diet with α-lipoic acid had no significant effect on the final body weight in ZL, but it prevented the increase in body weight in ZDF rats. As shown in Figure 2A, ZDF rats showed a higher ratio of epididymal fat/body weight (*p* < 0.01) in comparison to ZL rats. Alpha-lipoic acid had no effect on this ratio in ZL rats, but decreased it significantly in ZDF rats (*p* < 0.01; Figure 2A). In comparison to ZL rats, ZDF rats also exhibited a significant increase in the ratio of pancreas weight/body weight (*p* < 0.05; Figure 2B). Whereas the treatment with α-lipoic acid significantly enhanced (*p* < 0.05; Figure 2B) the pancreas weight/body weight in ZDF rats, it had no significant effect in ZL rats despite the trend.

### 3.3. Metabolic Parameters 

ZDF rats showed higher levels of plasma glucose (*p* < 0.01; Table 1) in comparison to ZL rats. The 6-week treatment with α-lipoic acid had no effect on plasma glucose in ZL rats and reduced it, although not significantly in ZDF rats. Likewise, plasma insulin levels and the HOMA index of insulin resistance were higher in ZDF rats in comparison to ZL rats (*p* < 0.01), but both metabolic parameters were not significantly improved by the treatment with α-lipoic acid (Table 1).

### 3.4. Oxidative Stress 

As shown in Figure 3, the basal superoxide anion production in the gastrocnemius muscle was significantly increased in ZDF rats (*p* < 0.01) in comparison to ZL rats. Supplementation of the diet with α-lipoic acid significantly reduced (*p* < 0.01) this increase in the gastrocnemius muscle of ZDF rats, but had no effect on the basal superoxide anion production in ZL rats.

### 3.5. Number and Size of Epididymal Adipocyte Cells

As shown in Figure 4A, the number of epididymal adipocyte cells was significantly (*p* < 0.01) decreased in ZDF in comparison to ZL rats. Treatment with α-lipoic acid had no significant effect on this parameter in ZL rats, yet it decreased it significantly (*p* < 0.01) in ZDF rats (Figure 4A). Inversely, the epididymal adipocyte cell size was significantly (*p* < 0.01) increased in ZDF in comparison to ZL rats and was further enhanced by α-lipoic acid (*p* < 0.01) in ZDF rats (Figure 4B). Thus, an increase in the size of adipocytes resulted in a decrease in their number per tissue section.

### 3.6. Epididymal UCP-1 Protein Staining

As shown in Figure 5, ZDF rats exhibited a significant (*p* < 0.05) decrease in epididymal UCP-1 protein staining in comparison to ZL rats. Treatment with α-lipoic acid led to a highly significant (*p* < 0.01) increase of this parameter in both ZL and ZDF rats so that the epididymal UCP-1 protein staining was significantly (*p* < 0.05) higher in ZDF rats treated with α-lipoic acid than in control ZL rats. This means that the antioxidant therapy not only normalized UCP-1 protein expression in ZDF rats, but enhanced it above normal values. 

## 4. Discussion

This study showed that ZDF rats exhibited hypertension and insulin resistance, accompanied by an increase in body weight and epididymal fat weight and a decrease in epididymal UCP-1 protein expression. Importantly, alpha-lipoic acid supplementation prevented the development of hypertension and reduced epididymal fat weight by a mechanism associated with the increase of epididymal UCP-1 protein expression. In addition, α-lipoic acid prevented the increase in basal O_2_•^−^ production in the gastrocnemius muscle and enhanced pancreatic weight in ZDF rats. 

Oxidative stress issued from an enhanced formation of ROS [30] is a fundamental event in the evolution of type 2 diabetes that may precede the occurrence of endothelial dysfunction and insulin resistance [31]. The ability of α-lipoic acid to combat the oxidative stress and improve systemic and skeletal muscle glucose utilization under conditions of insulin resistance has been substantiated in numerous models, including 3T3-L1 adipocytes [32], isolated rat skeletal muscle [20], in vivo rat models of insulin resistance [33,34,35,36] and type 2 diabetic patients [37,38]. Moreover, treatment with α-lipoic acid has been shown to improve metabolic parameters, the blood pressure, vascular reactivity and morphology of vessels already damaged by experimental type 1 diabetes [39].

Previous studies have shown that O_2_•^−^ production is increased in the aorta and mesenteric arteries of ZDF rats, and that α-lipoic acid attenuated the rise of O_2_•^−^ production and vascular dysfunction [6,40] and that of other oxidative stress markers, as shown by the diminution in the protein carbonyls in plantaris muscle and in the urine conjugated dienes [41]. Our recent study showed that α-lipoic acid treatment prevented the enhanced basal O_2_•^−^ production and NADPH oxidase activity in the thoracic aorta and the epididymal fat tissue in the same ZDF rats used for the present study [7]. This is in agreement with the present study showing that α-lipoic acid treatment also prevented the enhanced basal O_2_•^−^ production in the gastrocnemius muscle of ZDF rats. Similarly, antioxidant therapy with either α-lipoic acid, N-acetyl-L-cysteine or argan oil prevented the enhanced arterial blood pressure, O_2_•^−^ formation and NADPH oxidase activity in vascular, adipose and skeletal muscle tissues in a model of insulin resistance caused by chronic glucose feeding [5,19,27,33]. Similarly to the present study in ZDF rats, enhanced O_2_•^−^ formation resulting from increased NADPH oxidase activity was shown in the aorta of SHR [42], mineralocorticoid hypertensive rats [43] and angiotensin II hypertensive rats [44]. Collectively, these results suggest that vascular oxidative stress is implicated in the elevation of systolic arterial pressure in insulin resistance and in ZDF rats, as further supported herein by the antihypertensive effect of α-lipoic acid therapy.

In the present study, ZDF rats exhibited an increase in whole body and epididymal fat weight in association with a decrease in epididymal fat UCP-1 protein staining, which was prevented by six weeks treatment with α-lipoic acid. The increase of epididymal fat UCP-1 protein staining was beyond its normalization in ZDF rats. This is in keeping with a recent finding showing that dietary therapy with α-lipoic acid induces changes in gene expression in liver and adipose tissues, which ameliorate whole body lipid status in ZDF rats [45]. A recent study showed that the downregulation of UCP-1 mRNA and protein expression in brown adipose tissue in ZDF rats is reversed by six weeks of treatment with the potent antioxidant melatonin [46].Thus, one may suggest that α-lipoic acid exerts its anti-visceral obesity effects through its antioxidant properties, leading to the activation of UCP-1 protein expression at the adipose tissue level in ZDF rats.The beneficial impact of α-lipoic acid to decrease the body weight gain in ZDF rats is consistent with the reduction of NADPH oxidase hyperactivity in adipose tissue [7].

In the co-morbid conditions of obesity and type-2 diabetes, elevated adipocyte hypertrophy and enhanced inflammatory events are amongst the major detrimental factors that impair insulin signaling and deregulate glucose metabolism [47,48,49]. Indeed, in the present study, ZDF rats exhibited epididymal adipocyte hypertrophy in association with hyperglycemia, hyperinsulinemia and insulin resistance. This is in accordance with several studies, which have reported that adipocyte hypertrophy leads to reduced levels of adiponectin, an insulin-sensitizing and anti-inflammatory agent [48,49]. Interestingly, in the present study, six weeks of treatment with α-lipoic acid neither exerted significant protective effects on hyperglycemia, hyperinsulinemia and insulin resistance (HOMA index), nor prevented epididymal adipocyte hypertrophy in ZDF rats. Hence, one may suggest that the failure of α-lipoic acid to exert protective effects on glucose metabolism is explained partly by its inability to inhibit adipocyte hypertrophy and the subsequent elevation of pro-inflammatory mediators and reduction of adiponectin levels in ZDF rats. We also found that α-lipoic acid treatment resulted in an increase in pancreatic weight in ZDF rats, in accordance with the possibility that α-lipoic acid could preserve the impaired pancreatic insulin secretion without changing systemic insulin resistance [50]. Our study is supported by a report showing that dietary treatment with genistein, a potent antioxidant, did not improve glucose homeostasis in ZDF rats [51]. Thus, the present study indicates that factors other than oxidative stress and UCP-1 downregulation contribute to glucose metabolic alteration in ZDF rats.

## 5. Conclusions

Our data suggest that α-lipoic acid supplementation prevented the development of hypertension in ZDF rats, presumably through its antioxidant properties. Alpha-lipoic acid appears to exert its anti-visceral obesity effects through its antioxidant properties and by over-expression of the UCP-1 protein in the adipose tissue of ZDF rats. The failure of the α-lipoic acid treatment to impact on insulin resistance, hyperglycemia and hyperinsulinemia indicates that factors other than oxidative stress and UCP-1 downregulation contribute to the glucose metabolic disorders in ZDF rats. Hence, nutritional therapy with α-lipoic acid or its intake through consumable vegetables enriched with this compound may present benefits in the prevention of hypertension and obesity in type 2 diabetes.

## Figures and Tables

**Figure 1 antioxidants-08-00648-f001:**
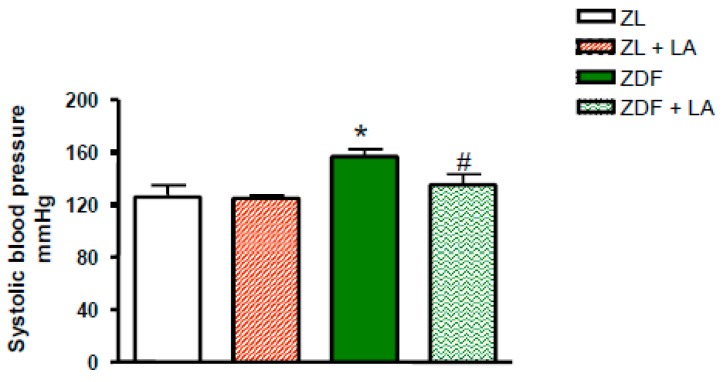
Effects of α-lipoic acid (LA) supplementation or standard diet on systolic blood pressure expressed in mmHg. Values are mean ± standard error of the mean (SEM) of 8 rats per group. * *p* < 0.05 vs. Zucker lean rats (ZL), ^#^
*p* < 0.05 vs. Zucker Diabetic Fatty rats (ZDF).

**Figure 2 antioxidants-08-00648-f002:**
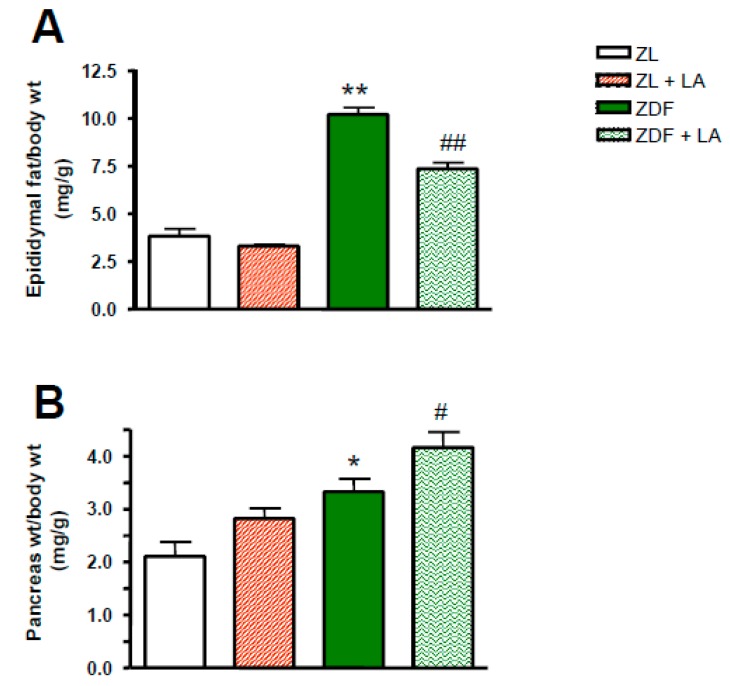
Effects of α-lipoic acid (LA) supplementation or standard diet on: (**A**) Epididymal fat/body weight expressed in mg/g; (**B**) Pancreas weight/body weight expressed in mg/g. Values are mean ± SEM of 8 rats per group. * *p* < 0.05, ** *p* < 0.01 vs. Zucker lean rats (ZL), ^#^
*p* < 0.05, ^##^
*p* < 0.01 vs. Zucker Diabetic Fatty rats (ZDF).

**Figure 3 antioxidants-08-00648-f003:**
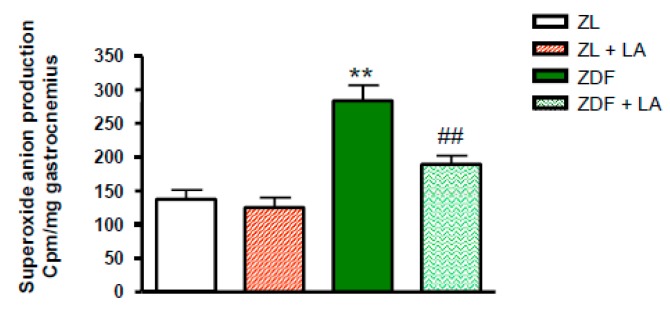
Effects of α-lipoic acid (LA) supplementation or standard diet on superoxide anion production expressed in cpm/mg of gastrocnemius muscle. Values are mean ± SEM of 8 rats per group. ** *p* < 0.01 vs. Zucker lean rats (ZL), ^##^
*p* < 0.01 vs. Zucker Diabetic Fatty rats (ZDF).

**Figure 4 antioxidants-08-00648-f004:**
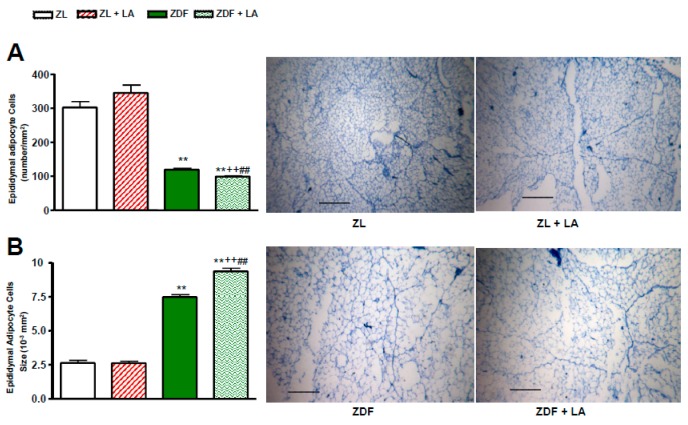
Effects of α-lipoic acid (LA) supplementation or standard diet on: (**A**) Epididymal adipocyte cell number expressed in number/mm^2^; (**B**) epididymal adipocyte cell size expressed in10^–3^ mm^2^.Values are mean ± SEM of 5–6 rats per group.** *p* < 0.01 vs. Zucker lean rats (ZL),^++^
*p* < 0.01 vs. Zucker Lean rats (ZL) treated with LA,^##^
*p* < 0.01 vs. Zucker Diabetic Fatty rats (ZDF). Histology of the adipose tissue pad is also shown for each group. Scale bar = 0.5 mm.

**Figure 5 antioxidants-08-00648-f005:**
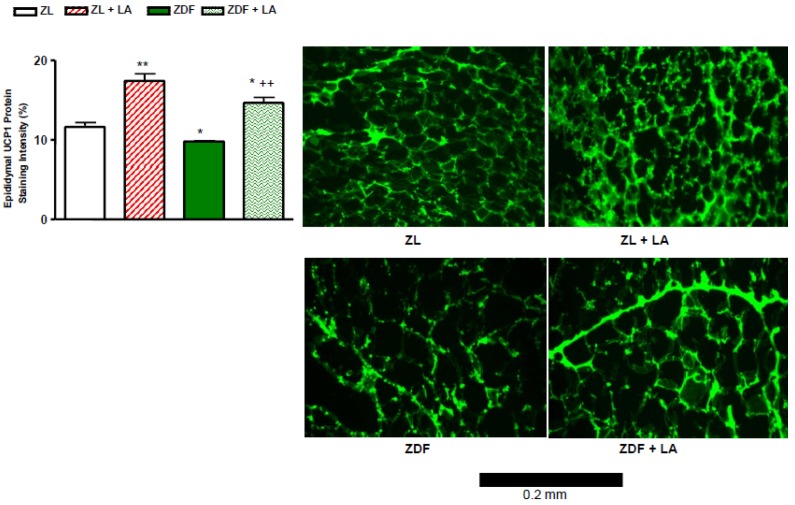
Effects of α-lipoic acid (LA) supplementation or standard diet on epididymal uncoupled protein-1 (UCP-1) staining intensity (%). Values are mean ± SEM of 5–6 rats per group.* *p* < 0.05, ** *p* < 0.01 vs. Zucker lean rats (ZL), ^++^
*p* < 0.01 vs. Zucker Diabetic Fatty rats (ZDF). Microphotograph of immunostaining of UCP-1 in the adipose tissue is also shown for each group. Scale bar = 0.2 mm.

**Table 1 antioxidants-08-00648-t001:** Effects of α-lipoic acid (LA) supplementation on final body weight, plasma levels of glucose and insulin, the index of insulin resistance (HOMA) in Zucker diabetic fatty (ZDF) and Zucker lean (ZL) rats.

Metabolic Parameters	ZL	ZL + LA	ZDF	ZDF + LA
Final body weight(g)	312 ± 6	305 ± 6	365 ± 9 **	326 ± 5 ^##^
Plasma glucose (mmol/L)	5.8 ± 0.4	5.3 ± 0.3	16.8 ± 1.8 **	12.9 ± 1.7 **
Plasma insulin (pmol/L)	471.5 ± 36.5	520.6 ± 130.2	2098.9 ± 313.5 **	2087.0 ± 253.9 **
HOMA	18.0 ± 1.9	17.5 ± 4.3	271.9 ± 30.9 **	182.4 ± 50.9 **

Results presented in this table are from our previous study in the same rats [7]. Data are means ± SEM of 8 rats per group. ** *p* < 0.01 vs. Zucker lean rats (ZL), ^##^
*p* < 0.01 vs. Zucker Diabetic Fatty rats (ZDF).

## Data Availability

The data used to support the findings of this study are available from the corresponding author upon request.

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
