# Peer review of "Beneficial Effects of Alpha-Lipoic Acid on Hypertension, Visceral Obesity, UCP-1 Expression and Oxidative Stress in Zucker Diabetic Fatty Rats"

_antioxidants, 2019, doi:10.3390/antiox8120648_

Round 1

Reviewer 1 Report

Manuscript by El Midaoui describes the effect of lipoic acid supplementation on several parameters of obese rats (ZDF). There are several points in the study that were unclear and reduced overall enthusiasm.  

1) It is a very unclear protocol for the treatment. Seems like LA supplementation started at week 6 and ended at week 12. It is unclear how an equal dose of lipoic acid was administered with diet, thus, it is hard to compare the effects of LA on WT and ZDF rats as they consume different amounts of food. Moreover, no data is provided for week 6 in regards to weight, glucose levels, pressure, etc, to understand whether the LA effect is protective or reversal. 

2) Superoxide was measured by chemiluminescence. No data provided if this effect was due to the antioxidant nature of LA or because of decreased expression of superoxide producing enzymes or an increase in superoxide dismutases. Is mitochondria the major source of superoxide or cytosol or membrane? Superoxide does not react with LA directly. Thus, it will be more important to check hydrogen peroxide levels. Endpoints of higher oxidative stress were not assessed (lipid peroxidation, carbonylation, nitration, dityrosine formation, oxidized glutathione).

3) LA is a co-factor of several enzymes. Metabolic changes could be explained by increased metabolism in the TCA cycle. Pyruvate dehydrogenase (PDH) activity should be assessed.

4) Superoxide was measured in muscle tissue, why UCP1 was not stained in the same tissue? Or authors should provide the measurement of superoxide in fat tissue. There are several uncoupler proteins in the mitochondria, the choice of UCP1 is unclear as well as the overall choice of uncouplers. The authors did not present evidence that mitochondria are compromised in ZDF rats.

5) The effect of LA in diabetes is well-reported and this study does not add any significant improvement in mechanistic understanding of its action.

Author Response

Reviewer 1

Manuscript by El Midaoui describes the effect of lipoic acid supplementation on several parameters of obese rats (ZDF). There are several points in the study that were unclear and reduced overall enthusiasm.  

1) It is a very unclear protocol for the treatment. Seems like LA supplementation started at week 6 and ended at week 12. It is unclear how an equal dose of lipoic acid was administered with diet, thus, it is hard to compare the effects of LA on WT and ZDF rats as they consume different amounts of food. Moreover, no data is provided for week 6 in regards to weight, glucose levels, pressure, etc, to understand whether the LA effect is protective or reversal. 

1- Yes, LA supplementation started at week 6 and ended at week 12. This is better clarified in the first paragraph of p. 3. We did not measure daily the quantity of food consumed although we observed no difference in the food intake between the standard diet and the LA supplementation diet. Because comparison was made between the two diets, it takes into account the polyphagia phenotype of ZDF. Thus, the difference between ZDF and ZL in the quantity of food intake is less determinant. The impact of LA supplementation diet was seen mainly in ZDF and was not uniform in all parameters, disclosing differential impact of the oxidative stress on them.

In this well-established animal model of metabolic syndrome (ZDF), it is noteworthy that this animal model develops features of metabolic syndrome with age. We decided to start the treatment of animals at 6 weeks, the age in which ZDF rats did not yet display the metabolic syndrome characteristics (Sato et al, Diabetes 59: 1092–1100, 2010). This is the reason why we did not collect data at 6 weeks. Our study is a preventive intervention; we performed the treatment of ZDF and their control counterparts ZL in parallel with the same diets. We found that LA prevented the development of arterial hypertension and exerted anti-visceral obesity in this model of metabolic syndrome. These clarifications are now included in the methods section at page 3, lines 92-93 and lines 95-96 of the revised manuscript.

2) Superoxide was measured by chemiluminescence. No data provided if this effect was due to the antioxidant nature of LA or because of decreased expression of superoxide producing enzymes or an increase in superoxide dismutases. Is mitochondria the major source of superoxide or cytosol or membrane? Superoxide does not react with LA directly. Thus, it will be more important to check hydrogen peroxide levels. Endpoints of higher oxidative stress were not assessed (lipid peroxidation, carbonylation, nitration, dityrosine formation, oxidized glutathione).

2- The reactive oxygen species production was evaluated by the lucigenin chemiluminescence method, a well-validated technique for measurement of cells superoxide anion (O2•−) production [7, 28]. The antioxidant nature of LA and its mechanism are well known and have been described in several reviews (see ref 16, Rochette et al.). LA is a free-radical scavenger and we reported in this study and in the previous one (ref 7) a marked reduction of superoxide in adipose tissue, skeletal muscle and vessel (P. 8, lines 242-246). LA is also converted to DHLA, which displays strong antioxidant activity (p. 2, lines 63-70). Moreover, superoxide is the main product of NADPH oxidase, which is upregulated in ZDF and other models as discussed on p. 8. Importantly, superoxide combines with NO to yield peroxynitrite that causes nitrosylation of proteins and enzymes, some of which are indeed involved in the oxidative stress balance and defence. Superoxide is located upstream to the cascade of oxidative molecules and is a reliable marker of the oxidative stress.

3) LA is a co-factor of several enzymes. Metabolic changes could be explained by increased metabolism in the TCA cycle. Pyruvate dehydrogenase (PDH) activity should be assessed.

3-The reviewer is raising an interesting issue. Pyruvate dehydrogenase (PDH) activity assessment would be informative. Unfortunately, the tissues used for this study are exhausted and no longer available. We will measure PDH activity in our future studies.

4) Superoxide was measured in muscle tissue, why UCP1 was not stained in the same tissue?Or authors should provide the measurement of superoxide in fat tissue. There are several uncoupler proteins in the mitochondria, the choice of UCP1 is unclear as well as the overall choice of uncouplers. The authors did not present evidence that mitochondria are compromised in ZDF rats.

4-We have published recently that lipoic acid prevented the increase of superoxide anion production in the epididymal fat tissue [7] in the same ZDF rats used for the present study. These clarifications are now clearly indicated in p. 8, lines 242-245 of the revised manuscript.

As mentioned in the Introduction, developing strategies to enhance the brown phenotype in white adipocytes to induce thermogenic activation has been proposed to combat obesity and type 2 diabetes [9,10]. Moreover, studies have shown that anti-diabetic peroxisome proliferator-activated receptor gamma (PPARγ) agonists induce a white-to-brown fat conversion andthermogenic gene expression through activation of UCP-1 [11-13].  In addition, Lapa et al 2017 (Sci Rep. 2017;7(1):16795) have reported that 13-weeks ZDF rats  presented ubiquitous white adipose-like tissue, showing large unilocular lipid droplets with marked decrease in immunostaining mitochondrial concentrations as well as in immunostaining and protein expression of UCP-1. These findings provide the rational for the choice of UCP-1 in our study. These clarifications are now included in page 2, lines 55-58 of the revised manuscript.

5) The effect of LA in diabetes is well-reported and this study does not add any significant improvement in mechanistic understanding of its action.

5-This study has shown that lipoic acid exerts anti-visceral obesity effect through induction of UCP-1 expression and its antioxidant properties at the adipose tissue level in ZDF. The finding that LA failed to ameliorate hyperglycemia, hyperinsulinemia and insulin resistance also shows that the oxidative stress is unlikely an important mechanism in glucose metabolic disorders in this model (ZDF).

N.B. Authors are most grateful to the reviewers for their helpful and valuable comments and suggestions that improved the outcome of this manuscript and for the perspective ideas of future studies.

Reviewer 2 Report

In this study, Midaoui et al evaluated the effects of dietary supplementation of α-lipoic acid (LA) on hypertension, insulin resistance, the increase in superoxide anion production in skeletal muscle, adiposity and the alteration in visceral UCP-1 protein expression in zucker diabetic fatty (ZDF) rat model. Based on the results, the authors concluded that LA supplementation can prevent the development of hypertension through its antioxidant properties. They suggested that the antioxidant properties and UCP-1 over-expression in adipose tissue may derive the anti-visceral obesity effects. The manuscript is basically well written and it will provide important information for the researchers in this study field. To make the findings more convincing, the authors should consider addressing the following concerns.

The authors concluded the antioxidant properties of LA by only the results from superoxide anion production in gastrocnemius muscle in ZDF rat model. To confirm whether the antioxidant properties of LA are really exerted in the animal model, other oxidative stress parameters, such as lipid and protein peroxidation, should be evaluated in the blood (or other organs). 

Author Response

Reviewer 2

In this study, Midaoui et al evaluated the effects of dietary supplementation of α-lipoic acid (LA) on hypertension, insulin resistance, the increase in superoxide anion production in skeletal muscle, adiposity and the alteration in visceral UCP-1 protein expression in zucker diabetic fatty (ZDF) rat model. Based on the results, the authors concluded that LA supplementation can prevent the development of hypertension through its antioxidant properties. They suggested that the antioxidant properties and UCP-1 over-expression in adipose tissue may derive the anti-visceral obesity effects. The manuscript is basically well written and it will provide important information for the researchers in this study field. To make the findings more convincing, the authors should consider addressing the following concerns.

The authors concluded the antioxidant properties of LA by only the results from superoxide anion production in gastrocnemius muscle in ZDF rat model. To confirm whether the antioxidant properties of LA are really exerted in the animal model, other oxidative stress parameters, such as lipid and protein peroxidation, should be evaluated in the blood (or other organs). 

We did not evaluate lipid or protein peroxidation as oxidative stress markers. However, in the present study, the reactive oxygen species production was evaluated by the lucigenin chemiluminescence method, a well-validated technique for measurement of cells superoxide anion (O2•−) production [28, 7]. Moreover, beside our evaluation of basal superoxide anion production in gastrocnemius skeletal muscle, the target organ for insulin sensitivity, we have evaluated the basal superoxide anion production as well NADPH oxidase activity, the two principal oxidative stress parameters, in the aortic tissue (a representative vessel for vascular abnormality in blood pressure) and in visceral adipose tissue in the same 12 weeks ZDF rats used for the present study (see ref 7). Indeed, we have reported recently that the supplemented diet with lipoic acid prevented the increase in the basal superoxide anion production and the NADPH oxidase activity in the aorta and the epididymal fat tissue in the same ZDF rats[7]. These clarifications are now clearly indicated in p. 8, lines 242-245 of the revised manuscript.

N.B. Authors are most grateful to the reviewers for their helpful and valuable comments and suggestions that improved the outcome of this manuscript and for the perspective ideas of future studies.

Reviewer 3 Report

Midaoui et al. prepared a valuable report on the "Beneficial Effects of Alpha-Lipoic Acid on Hypertension, Visceral Obesity, UCP-1 Expression and Oxidative Stress in Zucker Diabetic Fatty Rats". Methodology and study design are appropriate. Only issue would be statements similar to "These findings suggest that LA was efficacious in preventing the development of hypertension through its antioxidant properties." As the relationship between hypertension and antioxidant properties it is not straightforward, I would suggest stating "These findings suggest that LA was efficacious in preventing the development of hypertension that could be related to its antioxidant properties."

Author Response

Reviewer 3

Midaoui et al. prepared a valuable report on the "Beneficial Effects of Alpha-Lipoic Acid on Hypertension, Visceral Obesity, UCP-1 Expression and Oxidative Stress in Zucker Diabetic Fatty Rats". Methodology and study design are appropriate. Only issue would be statements similar to "These findings suggest that LA was efficacious in preventing the development of hypertension through its antioxidant properties." As the relationship between hypertension and antioxidant properties it is not straightforward, I would suggest stating "These findings suggest that LA was efficacious in preventing the development of hypertension that could be related to its antioxidant properties."

This modification is now included in page 1, lines 27-28 and page 9, line 288 of the revised manuscript.

N.B. Authors are most grateful to the reviewers for their helpful and valuable comments and suggestions that improved the outcome of this manuscript and for the perspective ideas of future studies.